# AVLEN: Audio-Visual-Language Embodied Navigation in 3D Environments

**Sudipta Paul**[1*]        **Amit K. Roy-Chowdhury**[1]        **Anoop Cherian**[2*]
spaul007@ucr.edu        amitrc@ece.ucr.edu        cherian@merl.com
[1]University of California, Riverside        [2]Mitsubishi Electric Research Labs, Cambridge, MA

## Abstract

Recent years have seen embodied visual navigation advance in two distinct directions: (i) in equipping the AI agent to follow natural language instructions, and (ii) in making the navigable world multimodal, e.g., audio-visual navigation. However, the real world is not only multimodal, but also often complex, and thus in spite of these advances, agents still need to understand the uncertainty in their actions and seek instructions to navigate. To this end, we present AVLEN – an *interactive agent* for Audio-Visual-Language Embodied Navigation. Similar to audio-visual navigation tasks, the goal of our embodied agent is to localize an audio event via navigating the 3D visual world; however, the agent may also seek help from a human (oracle), where the assistance is provided in free-form natural language. To realize these abilities, AVLEN uses a multimodal hierarchical reinforcement learning backbone that learns: (a) high-level policies to choose either audio-cues for navigation or to query the oracle, and (b) lower-level policies to select navigation actions based on its audio-visual and language inputs. The policies are trained via rewarding for the success on the navigation task while minimizing the number of queries to the oracle. To empirically evaluate AVLEN, we present experiments on the SoundSpaces framework for semantic audio-visual navigation tasks. Our results show that equipping the agent to ask for help leads to a clear improvement in performance, especially in challenging cases, e.g., when the sound is unheard during training or in the presence of distractor sounds.

## 1   Introduction

Building embodied robotic agents that can harmoniously co-habit and assist humans has been one of the early dreams of AI. A recent incarnation of this dream has been in designing agents that are capable of autonomously navigating realistic virtual worlds for solving pre-defined tasks. For example, in vision-and-language navigation (VLN) tasks [2], the goal is for the AI agent to either navigate to a goal location following the instructions provided in natural language, or to explore the visual world seeking answers to a given natural language question [10, 35, 36]. Typical VLN agents are assumed deaf; i.e., they cannot hear any audio events in the scene – an unnatural restriction, especially when the agent is expected to operate in the real world. To address this shortcoming, SoundSpaces [6] reformulated the navigation task with the goal of localizing an audio source in the virtual scene; however without any language instructions for the agent to follow.

Real-world navigation is not only audio-visual, but also is often complex and stochastic, so the agent must inevitably seek a synergy between the audio, visual, and language modalities for successful navigation. Consider, for example, a robotic agent that needs to find where the *"thud of a falling person"* or the *"intermittent dripping sound of water"* is heard from. On the one hand, such a sound may not last long and may not be continuously audible, and thus the agent must use semantic

---

*Equal Contribution.

36th Conference on Neural Information Processing Systems (NeurIPS 2022).

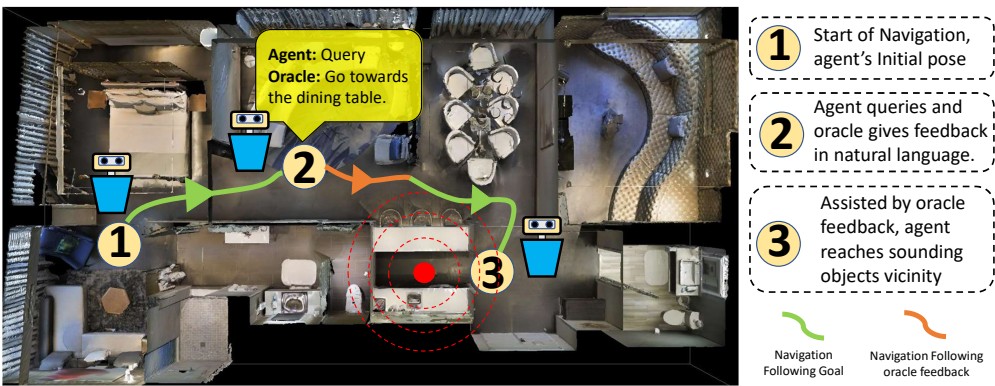

Figure 1: An illustration of our proposed AVLEN framework. The embodied agent starts navigating from location denoted ① guided by the audio-visual event at ③. At location ②, the learned policy for the agent decides to seek help from an oracle (e.g., because the audio stopped). The oracle provides a short natural language instruction for the agent to follow. The agent translates this instruction to produce a series of navigable steps to move towards the goal ③.

knowledge of the audio-visual modality [5] to reach the goal. On the other hand, such events need to be catered to timely and the agent should minimize the number of navigation mistakes it makes – a situation that can be efficiently dealt with if the agent can seek human help when it is uncertain of its navigation actions. Motivated by this insight, we present AVLEN – a first of its kind embodied navigation agent for localizing an audio source in a realistic visual world. Our agent not only learns to use the audio-visual cues to navigate to the audio source, but also learns to implicitly model its uncertainty in deciding the navigation steps and seeks help from an oracle for navigation instructions, where the instructions are provided in short natural language sentences. Figure 1 illustrates our task.

To implement AVLEN, we build on the realistic virtual navigation engine provided by the Matterport 3D simulator [2] and enriched with audio events via the SoundSpaces framework [6]. A key challenge in our setup is for the agent to decide when to query the oracle, and when to follow the audio-visual cues to reach the audio goal. Note that asking for help too many times may hurt agent autonomy (and is perhaps less preferred if the oracle is a human), while asking too few questions may make the agent explore the scene endlessly without reaching the goal. Further, note that we assume the navigation instruction provided to the agent is in natural language, and thus is often abstract and short (see Figure 1 above), making it difficult to be correctly translated to agent actions (as seen in VLN tasks [2]). Thus, the agent needs to learn the stochasticity involved in the guidance provided to it, as well as the uncertainty involved in the audio-visual cues, before selecting which modality to choose. To address these challenges, we propose a novel multimodal hierarchical options based deep reinforcement learning framework, consisting of learning a high-level policy to select which modality to use for navigation, among (i) the audio-visual cues, or (ii) natural language cues, and two lower-level policies that learn (i) to select navigation actions using the audio-visual features, or (ii) learns to transform natural language instructions to navigable actions conditioned on the audio-visual context. All the policies are end-to-end trainable and is trained offline. During inference, the agent uses its current state, the audio-visual cues, and the learned policies to decide if it needs oracle help or can continue navigation using the learned audio goal navigation policies.

Closely related to our motivations, a few recent works [9, 26, 27, 38] propose tasks involving interactions with an oracle for navigation. Specifically, in [9, 38] model uncertainty is used to decide when to query the oracle, where the uncertainty is either quantified in terms of the gap between the action prediction probabilities [9] to be less than a heuristically chosen threshold, or use manually-derived conditions to decide when an agent is lost in its navigation path [27]. In [26], the future actions of the policy of interest are required to be fully observed to identify when the agent is making mistakes and uses this information to decide when to query. Instead of resorting to heuristics, we propose a data-driven scheme to learn policies to decide when to query the oracle, these policies thus automatically learning the navigation uncertainty in the various modalities.

To empirically demonstrate the efficacy of our approach, we present extensive experiments on the language-augmented semantic audio-visual navigation (SAVi) task within the SoundSpaces framework for three very challenging scenarios, i.e., when: (i) the sound source is sporadic, however

familiar to the agent, (ii) sporadic but unheard of during training, and (iii) unheard and ambiguous due to the presence of simultaneous distractor sounds. Our results show clear benefits when the agent knows when to query and how to use the received instruction for navigation, as substantiated by improvements in success rate by nearly 3% when using the language instructions directly and by more than 10% when using the ground truth navigation actions after the query, even when the agent is constrained to trigger help only to a maximum of three times in a long navigation episode.

Before proceeding to detail our framework, we summarize below the main contributions of this paper.

- We are the first to unify and generalize audio-visual navigation with natural language instructions towards building a complete audio-visual-language embodied AI navigation interactive agent.
- We introduce a novel multimodal hierarchical reinforcement learning framework that jointly learns policies for the agent to decide: (i) when to query the oracle, (ii) how to navigate using audio-goal, and (iii) how to use the provided natural language instructions.
- Our approach shows state-of-the-art performances on the semantic audio-visual navigation dataset [5] with 85 large-scale real-world environments with a variety of semantic objects and their sounds, and under a variety of challenging acoustic settings.

## 2   Related Works

**Audio-Visual Navigation.** The SoundSpaces simulator, introduced in [6] to render realistic audio in 3D visual environments, pioneered research into the realm of audio-visual embodied navigation. The AudioGoal task in [6] consists of two sub-tasks, namely to localize an object: i) that sounds continuously throughout the navigation episode [6, 7] and ii) that sounds sporadically or can go mute at any time [5]. In this work, we go beyond this audio-visual setting, proposing a new task that equips the agent with the ability to use natural language – a setup that is realistic and practically useful.

**Instruction-Following Navigation.** There are recent works that attempt to solve the problem of navigation following instructions [2, 16, 17, 21, 25]. The instruction can be of many forms; e.g., structured commands [27], natural language sentences [2], goal images [22], or a combination of different modalities [26]). The task in vision-and-language navigation (VLN) for example is to execute free-form natural language instructions to reach a target location. To embody this task, several simulators have been used [2, 6, 29] to render real or photo-realistic images and perform agent navigation through a discrete graph [2, 8] or continuous environment [19]. One important aspect of vision and language navigation is to learn the correspondence between visual and textual information. To achieve this, [34] uses cross-modal attention to focus on the relevant parts of both the modalities. In [23] and [24], an additional module is used to estimate the progress, which is then used as a regularizer. In [13] and [31], augmented instruction-trajectory pairs is used to improve the VLN performance. In [37], long instructions are learned to be decomposed into shorter ones, executing them sequentially (via e.g., navigation). Recently, there are works using Transformer-based architectures for VLN [16, 25]. In [16], the BERT [11] architecture is used in a recurrent manner maintaining cross-modal state information. These works only consider the language-based navigation task. Different from these works, our proposed AVLEN framework solves vision-and-language navigation as a sub-task within the original semantic audio-visual navigation task [6].

**Interactive Navigation.** Recently, there have been works where an agent is allowed to interact with an oracle or a different agent, receiving feedback, and utilizing this information for navigation [9, 26, 27, 32]. The oracle instructions in these works are limited to either ground truth actions [9] or a direct mapping of a specific number of actions to consecutive phrases [27] is needed. Though [26] uses a fixed set of natural language instructions as the oracle feedback, it is coupled with the target image that the agent will face after completion of the sub-goal task. In Nguyen et al. [26], the agent needs to reach a specific location to query, which may be infeasible practically or sub-optimal if these locations are not chosen properly. Our approach differs fundamentally from these previous works in that we consider free-form natural language instructions and the agent can query the oracle from any navigable point in the environment, making our setup very natural and flexible.

## 3   Proposed Method

In this section, we will first formally define our task and our objective. This will be followed by details of our multimodal hierarchical reinforcement learning framework and our training setup.

**Problem Setup.** Consider an agent in a previously unseen 3D world navigable along a densely-sampled finite grid. At each vertex of this grid, the agent could potentially take one of a subset of actions from an action set $A = \{\text{stop}, \text{move\_forward}, \text{turn\_right}, \text{turn\_left}\}$. Further, the agent is assumed to be equipped with sensors for audio-visual perception via a binaural microphone and an ego-centric RGBD camera. The task of the agent in AVLEN is to navigate the grid from its starting location to find the location of an object that produces a sound (*AudioGoal*), where the sound is assumed to be produced by a static object and is semantically unique, however can be unfamiliar, sporadic, or ambiguous (due to distractors). We assume the agent calls the stop action only at the *AudioGoal* that terminates the episode. In contrast to the task in SoundSpaces, an AVLEN agent is also equipped with a language interface to invoke a *query* to an *oracle* under a budget (e.g., a limit on the maximum number of such queries). The oracle responds to the query of the agent via providing a natural language short navigation instruction; this instruction describing (in natural language) an initial *segment* along the shortest path trajectory from the current location of the agent to the goal. For example, for a navigation trajectory given by the actions $\langle\text{move\_forward}, \text{turn\_right}, \text{turn\_right}, \text{move\_forward}, \text{turn\_left}\rangle$, the corresponding language instruction provided by the oracle to the agent could be *"go around the sofa and turn to the door"*. As is clear, to use this instruction to produce navigable actions, the agent must learn to associate the language constructs with objects in the scene and their spatial relations, as well as their connection with the nodes in the navigation grid. Further, given the limited budget to make queries, the agent must learn to balance between when to invoke the query and when to navigate using its audio-visual cues. In the following, we present a multimodal hierarchical options approach to solve these challenges in a deep reinforcement learning framework.

**Problem Formulation.** We formulate the AVLEN task as a partially-observable Markov decision process (POMDP) characterized by the tuple $(\mathcal{S}, \mathcal{A}, T, R, \mathcal{O}, P, \mathcal{V}, \gamma)$, where $\mathcal{S}$ represents the set of agent states, $\mathcal{A} = A \cup \{\text{query}\}$ with the navigation actions $A$ defined above combined with an action to query the oracle, $T(s'|s, a)$ is the transition probability for mapping a state-action pair $(s, a)$ to a state $s'$, $R(s, a)$ is the immediate reward for the state-action pair, $\mathcal{O}$ represents a set of environment observations $o$, $P(o|s', a)$ captures the probability of observing $o \in \mathcal{O}$ in a new state $s'$ after taking action $a$, and $\gamma \in [0, 1]$ is the reward discount factor for long-horizon trajectories. Our POMDP also incorporates a language vocabulary $\mathcal{V}$ consisting of a dictionary of words that the oracle uses to produce the natural language instruction. As our environment is only partially-observable, the agent may not have information regarding its exact state, instead maintains a belief distribution $b$ over $\mathcal{S}$ as an estimate of its current state. Using this belief distribution, the expected reward for taking an action $a$ at belief state $b$ can be written as $R'(b, a) = \sum_{s \in \mathcal{S}} b(s) R(s, a)$. With this notation, the objective of the agent in this work is to learn a policy $\pi : \mathbb{R}^{|\mathcal{S}|} \times \mathcal{A} \to [0, 1]$ that maximizes the expected return defined by the value function $V^\pi$, while minimizing the number of queries made to the oracle; i.e.,

$$\arg\max_\pi V^\pi(b_0) \text{ where } V^\pi(b) = \mathbb{E}\left[\sum_{i=0}^{\infty} \gamma^i \left(R'(b_{t+i}, a_{t+i}) - \zeta(t+i)\mathbb{I}(a_{t+i} = \text{query})\right) | b_t = b, \pi\right],$$
(1)

where $\mathbb{I}$ denotes the indicator function, and the updated belief $b_{t+1} = \text{update}(o_{t+1}, b_t, a_t)$ defined for state $s'$ as $b_{t+1}(s') = \eta P(o_{t+1}|s', a_t) \sum_{s \in \mathcal{S}} b_t(s) T(s'|s, a_t)$ for a normalization factor $\eta > 0$. The function $\zeta : \mathbb{R}_+ \to \mathbb{R}$ produces a score balancing between the frequency of queries and the expected return from navigation.

At any time step $t$, the agent (in belief state $b_t$) receives an observation $o_{t+1} \in \mathcal{O}$ from the environment and selects an action $a_t$ according to a learned policy $\pi$; this action transitioning the agent to the new belief state $b_{t+1}$ as per the transition function $T'(b_{t+1}|a_t, b_t) = \sum_{o \in \mathcal{O}} \mathbb{I}(b_{t+1} = \text{update}(o, b_t, a_t)) P(o|s_t, a_t)$ while receiving an immediate reward $R'(b_t, a_t)$. As the navigation state space $\mathcal{S}$ of our agent is enormous, keeping a belief distribution on all states might be computationally infeasible. Instead, similar to [5], we keep a history of past $K$ observations in a memory module $M$, where an observation $o_t$ at time step $t$ is encoded via the tuple $e_t^o = (F_t^V, F_t^B, F_{t-1}^A, p_t)$ comprising neural embeddings of egocentric visual observation (RGB and depth) represented by $F_t^V$, the binaural audio waveform of the *AudioGoal* heard by the agent represented as a two channel spectrogram $F_t^B$, and the previous action taken $F_{t-1}^A$, alongside the pose of the agent $p_t$ with respect to its starting pose (consisting of the 3 spatial coordinates and the yaw angle). The memory $M$ is initialized to an empty set at the beginning of an episode, and at a time step $t$, is updated as $M = \{e_i^o : i = \max\{0, t - K\}, \dots, t\}$. Apart from these embeddings, AVLEN also incorporates a

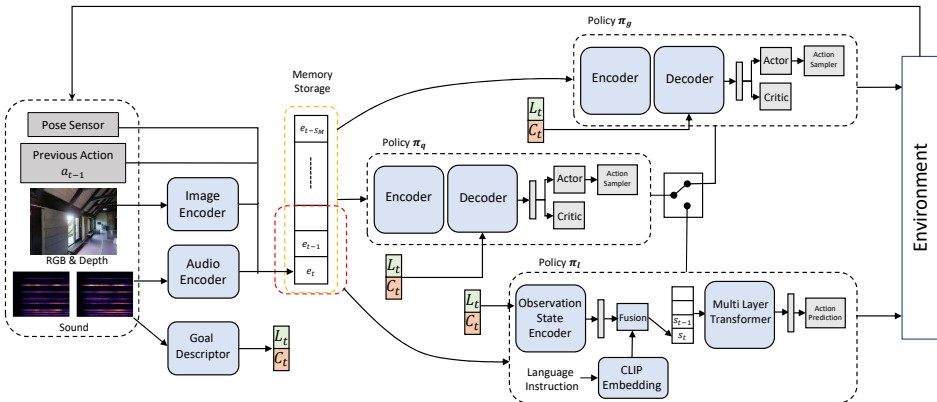

Figure 2: Architecture of our AVLEN pipeline. We show the two-level hierarchical RL policies that the model learns (offline), as well as the various input modalities and the control flow.

goal estimation network $f_g$ characterized by a convolutional neural network that produces a step-wise estimate $\hat{g}_t = f_g(F_t^B)$ of the sounding *AudioGoal*; $\hat{g}_t$ consisting of: (i) the (x, y) goal location estimate $L_t$ from the current pose of the agent, and (ii) the goal category estimate $c_t \in \mathbb{R}^C$ for $C$ semantic sounding object classes. The agent updates the current goal estimate combining the previous estimates as $g_t = \lambda \hat{g}_t + (1 - \lambda) f_p(g_{t-1}, \Delta p_t)$ where $f_p$ is a linear transformation of $g_{t-1}$ using the pose change $\Delta p_t$. We use $\lambda = 0.5$, unless the sound is inaudible in which case it is set to zero. We will use $g \in G \subset \mathbb{R}^{C+2}$ to denote the space of goal estimates.

**Multimodal Hierarchical Deep Reinforcement Learning.** As is clear, the diverse input modalities used in AVLEN have distinctly varied levels of semantic granularity, and thus a single monolithic end-to-end RL policy for navigation might be sub-optimal. For example, the natural language navigation instructions received from the oracle might comprise rich details for navigating the scene that the agent need not have to resort to any of the audio-visual inputs for say $\nu > 1$ steps. However, there is a budget on such queries and thus the agent must know when the query needs to be initiated (e.g., when the agent repeats sub-trajectories). Further, from a practical sense, each modality might involve different neural network architectures for processing, can have their own strategies for (pre-)training, involve distinct inductive baises, or incorporate heterogeneous navigation uncertainties.

All the above challenges naturally suggest to abstract the policy learning in the context of *hierarchical options* semi-Markov framework [3, 20, 30] consisting of low-level options corresponding to the navigation using either the *AudioGoal* or the language model, and a high-level policy to select among the options. More formally, an *option* is a triplet consisting of a respective policy $\xi$, a termination condition, and a set of belief states in which the option is valid. In our context, we assume a *multi-time* option policy for language-based navigation spanning $\nu$ navigation steps[2] and a *primitive* policy [30] for *AudioGoal*. We also assume these options may be invoked independent of the agent state. Suppose $\pi_q : \mathbb{R}^{|\mathcal{S}| \times |M|} \times G \times \{\text{query}\} \to [0, 1]$ represent the high-level policy deciding whether to query the oracle or not, using the current belief, the history $M$, and the goal estimate $g$. Further, let the two lower-level policies be: (i) $\pi_g : \mathbb{R}^{|\mathcal{S}| \times |M|} \times G \times A \to [0, 1]$, that is tasked with choosing the navigation actions based on the audio-visual features, and (ii) $\pi_\ell : \mathbb{R}^{|\mathcal{S}| \times \nu} \times \mathcal{V}^N \times G \times A \to [0, 1]$, that navigates based on the received natural language instruction formed using $N$ words from the vocabulary $\mathcal{V}$, assuming $\nu$ steps are taken after each such query. Let $R'_g$ and $R'_\ell$ denote the rewards (as defined in (1)) corresponding to the $\pi_g$ and $\pi_\ell$ options, respectively, where we have the multi-time discounted cumulative reward (with penalty $\zeta$) for $R'_\ell(b_t, a_t) = \mathbb{E}\left(\sum_{i=t}^{t+\nu-1} \gamma^{i-t} R'(b_i, a_i) | \pi_q = \pi_\ell, a_i \in A\right) - \zeta(t)$, while $R'_g$ is, being a primitive option, as in (1) except that the actions are constrained to $A$. Then, we have the Bellman equation for using the options given by:

$$V^\pi(b) = \pi_q(\xi_g|b)\left[R'_g + \sum_{o' \in \mathcal{O}} P'(o'|b, \xi_g)V^\pi(b')\right] + \pi_q(\xi_\ell|b)\left[R'_\ell + \sum_{o' \in \mathcal{O}} P(o'|b, \xi_\ell)V^\pi(b')\right], \quad (2)$$

---

[2]Otherwise terminated if the stop action is called before the option policy is completed.

where $\xi_g$ and $\xi_\ell$ are shorthands for $\xi = \pi_g$ and $\xi = \pi_\ell$, respectively, and $\pi = \{\pi_q, \pi_g, \pi_\ell\}$. Further, $P'$ is the multi-time transition function given by: $P'(o'|b, \xi) = \sum_{j=1}^{\infty} \sum_{s'} \sum_s \gamma^j P(s', o', j|s, \xi) b(s)$, where with a slight abuse of notation, we assume $P(s', o', j)$ is the probability to observe $o'$ in $j$ steps using option $\xi$ [30]. Our objective is to find the policies $\pi$ that maximizes the value function in (2) for $V^\pi(b_0)$. Figure 2 shows a block diagram of the interplay between the various policies and architectural components. Note that by using such a two-stage policy, we assume that the top-level policy $\pi_q$ implicitly learns the uncertainty in the audio-visual and language inputs as well as the predictive uncertainty in the respective low-level options $\pi_g$ and $\pi_\ell$ for reaching the goal state. In the next subsections, we detail the neural architectures for each of these options policies.

**Navigation Using Audio Goal Policy, $\pi_g$.** Our policy network for $\pi_g$ follows an architecture similar to [5], consisting of a Transformer encoder-decoder model [33]. The encoder sub-module takes in the embedded features $e^o$ from the current observation as well as such features from history stored in the memory $M$, while the decoder module takes in the output of the encoder concatenated with the goal descriptor $g$ to produce a fixed dimensional feature vector, characterizing the current belief state $b$. An actor-critic network (consisting of a linear layer) then predicts an action distribution $\pi_g(b, .)$ and the value of this state. The agent then takes an action $a \sim \pi_g(b, .)$, takes a step, and receives a new observation. The goal descriptor network $f_g$ outputs the object category $c$ and the relative goal location estimation $L$. Following SAVi [5], we use off-policy category level predictions and an on-policy location estimator.

**Navigation Using Language Policy, $\pi_\ell$.** When an agent queries, it receives natural language instruction $\text{instr} \in \mathcal{V}^N$ from the oracle. Using $\text{instr}$ and the current observation $e_t^o = (F_t^V, F_t^B, F_{t-1}^A, p_t)$, our language-based navigation policy performs a sequence of actions $\langle a_t, a_{t+1}, \ldots, a_{t+\nu} \rangle$ as per $\pi_\ell$ option, where each $a_i \in A$. Specifically, for any step $\tau \in \langle t, \ldots, t + \nu - 1 \rangle$, $\pi_\ell$ first encodes $\{e_\tau^o, g_\tau\}$ using a Transformer encoder-decoder network $T_1$[3], the output of this Transformer is then concatenated with CLIP [28] embeddings of the instruction, and fused using a fully-connected layer $\text{FC}_1$. The output of this layer is then concatenated with previous belief embeddings using a second multi-layer Transformer encoder-decoder $T_2$ to produce the new belief state $b_\tau$, i.e.,

$$b_\tau = T_2\left(\text{FC}_1\left(T_1(e_\tau^o, g_\tau), \text{CLIP(instr)}\right), \{b'_\tau : t < \tau' < \tau\}\right) \text{ and } \pi_\ell(b_\tau, .) = \text{softmax}(\text{FC}_2(b_\tau)). \tag{3}$$

**Learning When-to-Query Policy, $\pi_q$.** As alluded to above, the $\pi_q$ policy decides when to query, i.e., when to use $\pi_\ell$. Instead of directly utilizing model uncertainty [9], we use the reinforcement learning framework to be train this policy in an end-to-end manner, guided by the rewards $\zeta$.

**Reward Design.** For the $\pi_g$ policy, we assign a reward of +1 to reduce the geodesic distance towards the goal, a +10 reward to complete an episode successfully, i.e., calling the stop action near the *AudioGoal*, and a penalty of -0.01 per time step to encourage efficiency. As for the $\pi_\ell$ policy, we set a negative reward each time the agent queries the oracle, denoted $\zeta_q$, as well as when the query is made within $\tau$ steps from previous query, denoted $\zeta_f$. If the (softly)-allowed number of queries is $K$, then our combined negative reward function is given by $\zeta_q + \zeta_f$, where

$$\zeta_q(k) = \begin{cases} \frac{k \times (r_{neg} + \exp(-\nu))}{\nu} & k < K \\ r_{neg} + \exp(-k) & k \geq K, \end{cases} \quad \text{and} \quad \zeta_f(j) = \begin{cases} \frac{r_f}{j} & 0 < j < \tau \\ 0 & \text{otherwise}, \end{cases} \tag{4}$$

where $r_{neg}$ is set to -1.2, and $r_f$ is set to -0.5. As a result, the agent learns when to interact with the oracle directly based on its current observation and history information. In the RL framework, the actor-critic model also predicts the value function of each state. Policy training is done using decentralized distributed proximal policy optimization (DD-PPO).

**Policy Training.** Learning $\pi_g$ uses a two-stage training; in the first stage, the memory $M$ is not used, while in the second stage, observation encoders are frozen, and the policy network is trained using both the current observation and the history in $M$. The training loss consists of (i) the value-function loss, (ii) policy network loss to estimate the actions correctly, and (iii) an entropy loss to encourage exploration. Our language-based navigation policy $\pi_\ell$ follows a two stage training as well. The first stage consists of an off-policy training. We re-purpose the fine-grained instructions provided by [15] to learn the language-based navigation policy $\pi_\ell$. The second stage consists of on-policy training. During roll outs in our hierarchical framework, as the agent interacts with the oracle and

---

[3]This is different Transformer from the one used for $\pi_g$, however taking $g_\tau$ as input to the decoder.

Table 1: Comparison of performances against state of the art in heard and unheard sound settings.

| | Feedback | Heard Sound | | | | | Unheard Sound | | | | |
|---|---|---|---|---|---|---|---|---|---|---|---|
| | | Success ↑ | SPL ↑ | SNA ↑ | DTG ↓ | SWS ↑ | Success ↑ | SPL ↑ | SNA ↑ | DTG ↓ | SWS ↑ |
| Random Nav. | ✗ | 1.4 | 3.5 | 1.2 | 17.0 | 1.4 | 1.4 | 3.5 | 1.2 | 17.0 | 1.4 |
| ObjectGoal RL | ✗ | 1.5 | 0.8 | 0.6 | 16.7 | 1.1 | 1.5 | 0.8 | 0.6 | 16.7 | 1.1 |
| Gan et al. [14] | ✗ | 29.3 | 23.7 | **23.0** | 11.3 | 14.4 | 15.9 | 12.3 | 11.6 | 12.7 | 8.0 |
| Chen et al. [6] | ✗ | 21.6 | 15.1 | 12.1 | 11.2 | 10.7 | 18.0 | 13.4 | 12.9 | 12.9 | 6.9 |
| AV-WaN [7] | ✗ | 20.9 | 16.8 | 16.2 | 10.3 | 8.3 | 17.2 | 13.2 | 12.7 | 11.0 | 6.9 |
| SMT[12]+Audio | ✗ | 22.0 | 16.8 | 16.0 | 12.4 | 8.7 | 16.7 | 11.9 | 10.0 | 12.1 | 8.5 |
| SAVi [5] | ✗ | 33.9 | 24.0 | 18.3 | 8.8 | 21.5 | 24.8 | 17.2 | 13.2 | 9.9 | 14.7 |
| **AVLEN** | Language | **36.1** | **24.6** | 19.7 | **8.5** | **23.1** | **26.2** | **17.6** | **14.2** | **9.2** | **15.8** |
| **AVLEN** | GT Actions | **48.2** | **34.3** | **26.7** | **7.5** | **36.0** | **36.7** | **24.1** | **18.7** | **8.3** | **26.6** |

Table 2: Comparisons in heard and unheard sound settings against varied query-triggering methods.

| | Feedback | Heard Sound | | | | | Unheard Sound | | | | |
|---|---|---|---|---|---|---|---|---|---|---|---|
| | | Success ↑ | SPL ↑ | SNA ↑ | DTG ↓ | SWS ↑ | Success ↑ | SPL ↑ | SNA ↑ | DTG ↓ | SWS ↑ |
| Random | Language | 32.5 | 21.1 | 16.1 | 8.93 | 21.8 | 23.5 | 14.8 | 11.5 | 9.9 | 14.3 |
| Uniform | Language | 33.2 | 22.4 | 17.8 | 9.1 | 22.0 | 22.1 | 14.6 | 11.5 | 9.8 | 13.3 |
| Model Uncertainty | Language | 34.2 | 24.0 | 19.5 | 8.7 | 20.5 | 24.9 | 16.1 | 13.5 | 9.3 | 15.2 |
| **AVLEN** | Language | **36.1** | **24.6** | **19.7** | **8.5** | **23.1** | **26.2** | **17.6** | **14.2** | **9.2** | **15.8** |

receives language instructions, we use these instructions with the shortest path trajectory towards the goal to finetune $\pi_\ell$. In both cases, it is trained with an imitation learning objective. Specifically, we allow the agent to navigate on the ground-truth trajectory by following teacher actions and calculate a cross-entropy loss for each action in each step by, given by $-\sum_t a_{t*} \log(\pi_\ell(b_t, a_t))$ where $a^*$ is the ground truth action and $\pi_\ell(b_t, a_t)$ is the action probability predicted by $\pi_\ell$.

**Generating Oracle Navigation Instructions.** The publicly available datasets for vision-and-language tasks contain a fixed number of route-and-instruction pairs at handpicked locations in the navigation grid. However, in our setup, a navigating agent can query an oracle at any point in the grid. To this end, we assume the oracle knows the shortest path trajectory s_path from the current agent location to the *AudioGoal*, and from which the oracle selects a segment consisting of $n$ observation-action pairs (we use $n = 4$ in our experiments), i.e., $\text{s\_path} = \langle (o_0, a_0), (o_1, a_1), \ldots, (o_{n-1}, a_{n-1}) \rangle$. With this assumption, we propose to mimic the oracle by a *speaker* model [13], which can generate a distribution of words $P^S(w | \text{s\_path})^4$. The observation and action pairs are sequentially encoded using an LSTM encoder, $\langle F_0^S, F_1^S, \ldots, F_n^S \rangle = \text{SpeakerEncoder}(\text{s\_path})$ and decoded by another LSTM predicting the next word in the instruction by: $w_t = \text{SpeakerDecoder}(w_{t-1}, \langle F_0^S, F_1^S, \ldots, F_n^S \rangle)$. The instruction generation model is trained using the available (instruction, trajectory) pairs from the VLN dataset [2]. We use cross entropy loss and teacher forcing during training.

## 4 Experiments and Results

**Dataset.** We use the SoundSpaces platform [6] for simulating the world in which our AVLEN agent conducts the navigation tasks. Powered by Matterport3D scans [4], SoundSpaces facilitates a realistic simulation of a potentially-complex 3D space navigable by the agent along a densely sampled grid with 1m grid-cell sides. The platform also provides access to panoramic ego-centric views of the scene in front of the agent both as RGB and as depth images, while also allowing the agent to hear realistic binaural audio of acoustic events in the 3D space. To benchmark our experiments, we use the semantic audio-visual navigation dataset from Chen et al. [5] built over SoundSpaces. This dataset consists of sounds from 21 semantic categories of objects that are visually present in the Matterport3D scans. The object-specific sounds are generated at the location of the Matterport3D objects. In each navigation episode, the duration of the sounds are variable and is normal-distributed with a mean 15s and deviation 9s, clipped for a minimum 5s and maximum 500s [5]. There are 0.5M/500/1000 episodes available in this dataset for train/val/test splits respectively from 85 Matterport3D scans.

**Evaluation Metrics.** Similar to [5], we use the following standard metrics for evaluating our navigation performances on this dataset: i) *success rate* for reaching the *AudioGoal*, ii) *success weighted by inverse path length* (SPL) [1], iii) *success weighted by inverse number of actions* (SNA) [7], iv) *average distance to goal* (DTG), and v) *success when silent* (SWS). SWS refers to the fraction of successful episodes when the agent reaches the goal after the end of the acoustic event.

---

[4]s_path is approximated in the discrete Room-to-Room [2] environment and then used to generate instruction.

Table 3: Comparisons against varied query-triggering methods with ground truth action as feedback.

| | Feedback | Heard Sound | | | | | Unheard Sound | | | | |
| | | Success ↑ | SPL ↑ | SNA ↑ | DTG ↓ | SWS ↑ | Success ↑ | SPL ↑ | SNA ↑ | DTG ↓ | SWS ↑ |
|---|---|---|---|---|---|---|---|---|---|---|---|
| Random | GT Actions | 39.8 | 30.0 | 24.5 | 7.6 | 23.3 | 29.6 | 22.1 | 18.4 | 8.2 | 16.3 |
| Uniform | GT Actions | 38.8 | 29.9 | 25.5 | 7.4 | 20.3 | 28.6 | 21.3 | 18.0 | **7.8** | 14.8 |
| Model Uncertainty | GT Actions | 41.3 | 30.6 | 24.8 | **7.3** | 26.3 | 31.4 | 22.7 | 18.4 | 8.2 | 19.3 |
| **AVLEN** | GT Actions | **48.2** | **34.3** | **26.7** | 7.5 | **36.0** | **36.7** | **24.1** | **18.7** | 8.3 | **26.6** |

Figure 3: (a) Comparison of AVLEN performances against baselines and when-to-query approaches in the *presence of distractor sound*, (b) Performance (SPL) comparison against varying the number of allowed queries, and (c) Distribution of queries triggered against the time steps in episodes.

| | Feedback | Success ↑ | SPL ↑ | SNA ↑ | DTG ↓ | SWS ↑ |
|---|---|---|---|---|---|---|
| Chen et al. [6] | ✗ | 4.0 | 2.4 | 2.0 | 14.7 | 2.3 |
| AV-WaN [7] | ✗ | 3.0 | 2.0 | 1.8 | 14.0 | 1.6 |
| SMT[12]+Audio | ✗ | 4.2 | 2.9 | 2.1 | 14.9 | 2.8 |
| SAVi [5] | ✗ | 11.8 | 7.4 | 5.0 | 13.1 | 8.4 |
| Random | Language | 11.6 | 6.6 | 4.8 | 12.9 | 7.8 |
| Uniform | Language | 11.6 | 6.8 | 5.1 | 13.3 | 7.7 |
| Model Uncertainty | Language | 12.4 | 6.7 | 5.0 | 12.8 | 8.4 |
| **AVLEN** | Language | **14.0** | **8.4** | **5.9** | **12.8** | **11.1** |
| **AVLEN** | GT Actions | **24.4** | **15.3** | **11.3** | **11.3** | **21.5** |

(a) Comparisons under distractors

(b) Query sensitivity

(c) Query distribution

**Implementation Details.** Similar to prior works, we use RGB and depth images, center-cropped to $64 \times 64$. The agent receives binaural audio clip as $65 \times 26$ spectrograms. The memory size for $\pi_g$ and $\pi_q$ is 150 and for $\pi_\ell$ is 3. All the experiments consider maximum $K = 3$ allowed queries (unless otherwise specified). For each query, the agent will take $\nu = 3$ navigation steps in the environment using the natural language instruction. We use a vocabulary with 1621 words. Training uses ADAM [18] with learning rate $2.5 \times 10^{-4}$. Refer to the Appendix for more details.

**Experimental Results and Analysis.** The main objective of our AVLEN agent in the semantic audio-visual navigation task is to navigate towards a sounding object in an unmapped 3D environment when the sound is sporadic. Since we are the first to integrate oracle interactions (in natural language) within this problem setting, we compare with existing state-of-the-art semantic audio-visual navigation approaches, namely Gan et al. [14], Chen et al. [6], AV-WaN [7], SMT [12] + Audio, and SAVi [5]. Following the protocol used in [5] and [6], we evaluate performance of the same trained model on two different sound settings: i) *heard sound*, in which the sounds used during test are heard by the agent during training, and ii) *unheard sound*, in which the train and test sets use distinct sounds. In both experimental settings, the test environments are unseen.

Table 1 provides the results of our experiments using heard and unheard sounds. The table shows that AVLEN (language) – which is our full model based on language feedback – shows a $+2.2\%$ and $+1.6\%$ absolute gain in success rate and success-when-silent (SWS) respectively, compared to the best performing baseline SAVi [5] for heard sound. Moreover, we obtain $1.4\%$ and $1.1\%$ absolute gain in success rate and SWS respectively for unheard sound compared to the next best method, SAVi. Our results clearly demonstrate that the agent is indeed able to use the short natural language instructions for improving the navigation.

A natural question to ask in this setting is: *Why are the improvements not so dramatic, given the agent is receiving guidance from an oracle?* Generally, navigation based on language instructions is a challenging task in itself ( [2, 25]) since language incorporates strong inductive biases and usually spans large vocabularies; as a result the action predictions can be extremely noisy, imprecise, and misguiding. However, the key for improved performance is to identify *when to query*. Our experiments show that AVLEN is able to identify when to query correctly (also see Table 2) and thus improve performance. To further substantiate this insight, we designed an experiment in which the agent is provided the ground truth (GT) navigation actions as feedback (instead of providing the corresponding language instruction) whenever a query is triggered. The results of this experiment – marked AVLEN (GT Actions) in Table 1 – clearly show an improvement in success rate by nearly $+15\%$ for heard sounds and $+12\%$ for unheard sounds, suggesting future work to consider improving language-based navigation policy $\pi_\ell$.

**Navigation Under Distractor Sounds.** Next, we consider the presence of distractor sounds while navigating towards an "unheard" sound source as provided in SAVi [5]. In this setting, the agent must know which sound its target is. Thus, a one hot encoding of the target is also provided as an input to the agent, if there are multiple sounds in the environment. The presence of distractor sounds

| | Feedback | Success ↑ | SPL ↑ | SNA ↑ | DTG ↓ | SWS ↑ |
|---|---|---|---|---|---|---|
| SAVi | ✗ | 33.9 | 24.0 | 18.3 | 8.8 | 21.5 |
| AVLEN (Glove + GRU) | Language | 36.1 | 24.6 | 19.5 | 8.5 | 23.3 |
| AVLEN (Glove + Transformer) | Language | 36.2 | 25.4 | 20.0 | 8.4 | 23.8 |
| AVLEN (CLIP+Transformer) | Language | 36.1 | 24.6 | 19.7 | 8.5 | 23.1 |
| AVLEN (CLIP + GRU) | Language | 37.7 | 25.5 | 19.9 | 8.5 | 25.1 |

Table 4: Comparisons in performance for different architectural choices for language-based policy $\pi_\ell$ in heard sound setting.

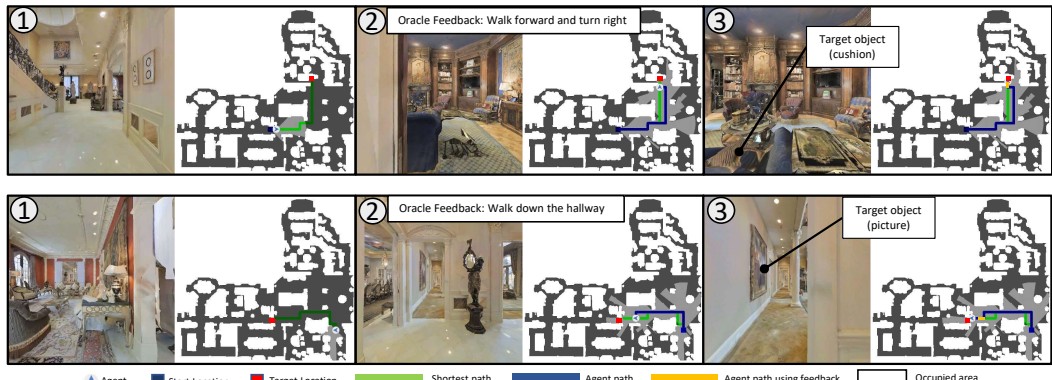

Figure 4: Two qualitative results from AVLEN's navigation trajectories. We show egocentric views and top down maps for three different viewpoints in agent's trajectory. The agent starts from ①, receives oracle help in ②, navigates to the goal in ③. In the top episode, agent receives directional information ('Walk forward and turn right'), whereas in the bottom episode, agent receives language instruction more grounded on the scene ('Walk down the hallway').

makes the navigation task even more challenging, and intuitively, the agent's ability to interact with the oracle could come useful. This insight is clearly reflected in Figure 3a, where AVLEN shows 2.2% and 2.7% higher success rate and SWS respectively compared to baseline approaches. Figure 3a shows that AVLEN outperforms other query procedures as well and the performance difference is more significant compared to when there is no distractor sound.

**Analyzing When-to-Query.** To evaluate if AVLEN is able to query at the appropriate moments, we designed potential baselines that one could consider for deciding when-to-query and compare their performances in Table 2. Specifically, the considered baselines are: i) *Uniform*: queries after every 15 steps, ii) *Random*: queries randomly within the first 50 steps of each episode, and iii) *Model Uncertainty (MU)* based on [9]: queries when the softmax action probabilities of the top two action predictions of $\pi_g$ have an absolute difference are less than a predefined threshold ($\leq 0.1$). Table 2 shows our results. Unsurprisingly, we see that Random and Uniform perform poorly, however using MU happen to be a strong baseline. However, MU is a computationally expensive baseline as it requires a forward pass on the $\pi_g$ model to decide when to query. Even so, we observe that compared to all the baselines, AVLEN shows better performance in all metrics. To further understand this, in Figure 3c we plot the distribution of the episode time step when the query is triggered for the unheard sound setting. As is clear, MU is more likely to query in the early stages of the episode, exhausting the budget, while AVLEN learns to distribute the queries throughout the steps, suggesting our hierarchical policy $\pi_q$ is learning to be conservative, and judicial in its task. Interestingly we also find reasonable overlap between the peaks of the two curves, suggesting that $\pi_q$ is considering the predictive uncertainty of $\pi_g$ implicitly.

*Is the AVLEN reward strategy effective to train the model to query appropriately?* To answer this question, we analyzed the effectiveness of our training reward strategy (i.e., to make the agent learn when to query) by comparing the performances of AVLEN with the other baseline querying methods (i.e.,*Random*, *Uniform*, and *Model Uncertainty*) when the oracle feedback provides the ground truth navigation actions, instead of language instructions. The results are provided in Table 3 and clearly show the superior performances of AVLEN against the alternatives suggesting that the proposed reward signal is effective for appropriately biasing the query selection policy.

*Are the improvements in performance from better Transformer and CLIP models?* To understand the architectural design choices (e.g., modules used in $\pi_\ell$) versus learning the appropriate policy to query (learning of $\pi_q$), we conducted an ablation study replacing the sub-modules in $\pi_\ell$ neural network with close alternatives. Specifically, we consider four architectural variants for $\pi_\ell$ for comparisons,

namely: (i) AVLEN (Glove + GRU), (ii) AVLEN (Glove + Transformer), (iii) AVLEN (CLIP + GRU), and (iv) AVLEN (CLIP + Transformer). We compare the performances in Table 4. Our results show that the improvements when using AVLEN is not due to CLIP context or Transformer representations alone, instead it is from our querying framework, that consistently performs better than the baseline SAVi model, irrespective of the various ablations.

**Sensitivity to Allowed Number of Queries, $\nu$.** To check the sensitivity AVLEN for different number of allowed queries (and thus the number of natural language instructions received), we consider $\nu \in \{0, 1, 2, 3, 4, 5\}$ and evaluate the performances. Figure 3b shows the SPL scores this experiment in the unheard sound setting. As is expected, increasing the number of queries leads to an increase in SPL for AVLEN, while alternatives e.g., Random drops quickly; the surprising behaviour of the latter is perhaps a mix of querying at times when the $\pi_g$ model is confident and the $\pi_\ell$ instructions being noisy.

**Qualitative Results.** Figure 4 provides snapshots from two example episodes of semantic audio-visual navigation using AVLEN. The sounding object is a 'cushion' in the first episode (first row) and a 'picture' in the second episode (second row). ② of both episodes shows the viewpoint where agent queries and receive natural language instructions. In the first row, agent receives directional information: 'Walk forward and turn right', while in the second row episode, the agent receives language instruction ground in the scene: 'Walk down the hallway'. In both cases, the agent uses the instruction to assist its navigation task and reach around the vicinity of target object ③. Please refer to the supplementary materials for more qualitative results.

## 5 Conclusions

The ability to interact with oracle/human using natural language instructions to solve difficult tasks is of great importance from a human-machine interaction standpoint. In this paper, we considered such a task in the context of audio-visual-language embodied navigation in a realistic virtual world, enabled by the SoundSpaces simulator. The agent, visualy navigating the scene to localize an audio goal, is also equipped with the possibility of asking an oracle for help. We modeled the problem as one of learning a multimodal hierarchical reinforcement learning policy, with a two-level policy model: higher-level policy to decide when to ask questions, and lower-level policies to either navigate using the audio-goal or follow the oracle instructions. We presented experiments using our proposed framework; our results show that using the proposed policy allows the agent achieve higher success rates on the semantic audio-visual navigation task, especially in cases when the navigation task is more difficult in presence of distractor sounds.

## 6 Limitations and Societal Impacts

**Limitations.** There are two key ingredients in our design that the performance of AVLEN is dependent on, namely (i) the need for a strong language-based navigation policy ($\pi_\ell$) to assist the semantic audio-visual navigation task, and (ii) a pre-trained *Speaker* model to generate language instructions so that AVLEN can cater to a query at any location in the 3D navigation grid. These modules bring in strong priors (from their pre-training) that in some of the cases the generated language instructions can be irrelevant, ambiguous, or incorrect, hampering the performance of the entire system. Further, this work uses only English language datasets and the MatterPort 3D dataset includes scans that mainly resemble North American houses, potentially causing an unintended model bias.

**Societal Impacts.** Our proposed AVLEN framework is a step towards building a human-robot cohesive ecosystem synergizing each other towards solving complex real-world problems. While, full autonomy of the robotic agents could be a important requirement for several tasks, we may not want an agent to take actions that it is unsure of, and our framework naturally enables such a setting. However, as noted in our experimental results, there seems to be a gap in how the language is interpreted by the agent when translating to its actions, and this could be a point of societal concern, even when there is the possibility of interactions with the humans.

**Acknowledgement.** SP worked on this problem as part of a MERL internship. SP and AR are partially supported by the US Office of Naval Research grant N000141912264 and the UC Multi-Campus Research Program through award #A21-0101-S003. The authors thank Mr. Xiulong Liu (UW) and Dr. Moitreya Chatterjee (MERL) for useful discussions.

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
