# OpenReview forum: "AVLEN: Audio-Visual-Language Embodied Navigation in 3D Environments"
_NeurIPS.cc/2022/Conference — NeurIPS 2022 Accept_

### Official Review · Reviewer_FS8x · 2022-07-13

**Rating:** 4
**Confidence:** 5
**Soundness:** 2 fair
**Presentation:** 3 good
**Contribution:** 2 fair

**Summary:**

The manuscript modifies the semantic audio-visual navigation task to incorporate oracle feedback, intermittently, throughout episodes. The approach leverages an options framework, in order to encourage the policy to learn when to query the oracle.

**Questions:**

Why is assuming access to an oracle considered a reasonable problem setting?

In order to more flexibly accommodate the penalties, why not formulate the problem as a constrained POMDP, with, among other signals, the queries as costs? Indeed, it would provide more control over the query budget. Can you discuss the tradeoff(s) in doing so?

**Limitations:**

See above — i.e., reward design, problem formulation, (lack of) ablations

**Strengths And Weaknesses:**

Strengths

The manuscript provides an options framework and extends the SAVi baseline to incorporate natural language.

The manuscript is reasonably well-written.

Weaknesses

Section 2: Consider other interaction paradigms — see some of the new embodied AI surveys, e.g., https://www.jair.org/index.php/jair/article/download/13646/26807/

Section 3: More analysis necessary: I would have liked to see a discussion that justifies that the reward signal is sufficiently expressive for appropriately biasing the high-level policy.

Section 3: Whereas predicating the reward design on distance-to-goal may have been essential to get started on applying RL frameworks to these embodied tasks, it is now important to move beyond this. Methods should pursue self-supervisory alternatives as reward signals, which would narrow the gap to practical real-world deployment.

Section 4: The manuscript is missing sufficient ablations, to confirm that the performance improvements are indeed resulting from the querying framework, as opposed to the more-expressive CLIP context, transformer representations, and VLN pre-training tasks; these are not present in the previous baselines. Despite the addition of those components, with unsupported contribution, the improvements are admittedly not so dramatic, as the manuscript self-admits.

---

> ### Author Response · Authors · 2022-08-02
> **Response to Reviewer FS8x**
>
> We thank the reviewer for the insightful questions, comments, and useful suggestions.
> - **See some of the new embodied AI surveys:**
> We will cite this survey and also will include related references.
> - **Discussion that justifies that the reward signal is expressive for biasing the policy:**
> Thanks for this careful point. To analyze this expressiveness of the language model, we compare AVLEN to other querying methods, including querying randomly, uniformly in time, and when the model uncertainty is high while receiving ground truth instructions after the query. We believe the superior performance that AVLEN showcases, in Table 2 and in the table below, compared to these alternatives is a clear indicator that the used reward signal is sufficiently expressive for appropriately biasing the high-level learned policy. Result for Unheard and Heard sounds are provided below, respectively.
>     |  Approach | Unheard| Heard|  Success | SPL | SNA | DTG | SWS |
>     |:-:|:-:|:-:|:-:|:-:|:-:|:-:|:-:|
>     |Uniform (GT)|&check; ||  28.6 | 21.3 | 18.0 | 7.77 | 14.8 |
>     |Random (GT)|&check;|| 29.6 | 22.1 | 18.4 | 8.19 | 16.3 |
>     |Model Uncertainty (GT) |&check; || 31.4 | 22.7 | 18.4 | 8.15 | 19.3 |
>     |AVLEN (GT)|&check;|| 36.7 | 24.1 | 18.7 | 8.29 | 26.6 |
>     |Uniform (GT)||&check;| 38.8 | 29.9 | 25.5 | 7.35 | 20.3 |
>     |Random (GT)||&check;| 29.6 | 22.1 | 18.4 | 8.19 | 16.3 |
>     |Model Uncertainty (GT)||&check;| 41.3 | 30.6 | 24.8 | 7.3 | 26.3 |
>     |AVLEN (GT)||&check;|  48.2 | 34.3 | 26.7 | 7.5 | 36.0 |
> - **Methods should pursue self-supervisory alternatives as reward signals.:**
> Thanks for this valuable suggestion. While we agree that methods should go beyond supervisory signals for solving tasks, note that our current work is scoped with the goal of integrating multimodal reasoning in a holistic setup towards an interactive agent for the navigation task, which we believe is the first attempt towards this goal. In that respect, our design choices were influenced by prior works, from which we did not want to deviate significantly at this stage. However, we sincerely value the suggestion of the reviewer and will use self-supervisory signals for better learning of the policies in the future revision of this work.
> - **The manuscript is missing sufficient ablations.:**
> We did miss to include these ablation studies in the main paper, which we rectify below. Specifically, we consider four setups for comparisons: (i) AVLEN (Glove + GRU),   (ii) AVLEN (Glove + Transformer), (iii)  AVLEN (CLIP + GRU) , and (iv) AVLEN (CLIP+Transformer). Our results are provided below. As is clear from these results, the improvements to AVLEN are not due to CLIP context or Transformer representations alone, instead it is from our querying framework that consistently performs better than the baseline SAVi model, irrespective of the various ablations. For Heard sound:
>     |Approach| Success | SPL | SNA | DTG | SWS |
>     | :-:|   :-:   | :-: | :-: | :-: | :-: |
>     |SAVi | 33.9 | 24.0 | 18.3 | 8.8 | 21.5 |
>     | AVLEN (Glove + GRU)   | 36.1 | 24.6 | 19.5 | 8.5 | 23.3 |
>     | AVLEN (Glove + Transformer)  |  36.2 | 25.4 | 20.0 | 8.4 | 23.8 |
>     | AVLEN (CLIP+Transformer)  |  36.1 | 24.6 |  19.7 |  8.5 |   23.1 |
>     | AVLEN (CLIP + GRU) | 37.7 | 25.5 | 19.9 | 8.5 | 25.1 |
> - **Why is assuming access to an oracle considered a reasonable problem setting?:**
> While, we do not expect the agent to access the oracle at all times, we believe having the potential to seek guidance is an important functionality that an agent should manifest. For example, for a surveillance agent, it may be possible that the agent may not be able to make decisions at certain times (e.g., it can listen to the drip of water but cannot exactly localize it), and thus having the oracle, which could be a human surveillance operator (similar to an elevator operator in a building) or another robotic agent, that provides instructions to precisely check locations where a water leak is possible, is important.  Further, using natural language instructions allows for seamless communication beyond robotic/devices, and includes humans within the navigation loop.
> - **Why not formulate the problem as a constrained POMDP?:**
> Thanks for this wonderful suggestion. We agree that using a constrained POMDP is a natural choice for our problem, where the budget constraints can be naturally included in the objective via the Langraigan dual. However, as far as we know such CPOMDPs within a large scale deep RL setting as ours is computationally too complex to train effectively. As a result, we sought alternatives using the options framework, that allows for flexible incorporation of various losses that are used in prior works, while also allows for including our new modules, such that the training of our architecture need not deviate much from publicly available implementations. We will note these factors in the revised paper, and we will consider CPOMDPs as important directions for future work.

---

### Official Review · Reviewer_ob8D · 2022-07-15

**Rating:** 7
**Confidence:** 4
**Soundness:** 3 good
**Presentation:** 3 good
**Contribution:** 3 good

**Summary:**

This work proposes a new audio-visual-language embodied navigation task, where the agent not only hears the audio from the sounding object but also is able to query the environment of the goal location. To deal with the multimodal navigation task, this paper propses a multimodal hierarchical policy that consists a high-level pocliy that decides which low-level policy and two low-level policies that either navigates with audio-visual cues or navigates with queried instructions. In the experiment, authors compare with the SOTA model in semantic auido-visual navigation and show that the proposed model outperforms the SOTA model.

**Questions:**

The trained oracle speaker seems to be useful for reproduce the results. Will you release the model and code?

In the supp video, it looks like a couple successful cases were because of the query of stop. Is it possible most the performance improvement comes from the stop query?

Do authors have an intution of why the performance improvement is not very big? It seems like the language instructions provide pretty strong guidance for navigation.


**Ethics Review Area:**

["I don’t know"]

**Limitations:**

Both the limitations and societal impact were discussed in the conclusion.

**Strengths And Weaknesses:**

The proposed task is interesting and adds one more modality to the audio-visual navigation task, although the ability to query is not very realistic in the real world.

The multimodal hierarchical model is also pretty novel in that the hierarchy is for in modality selection rather than the common goal prediction/actuation command hierarchy.

Using RxR dataset to train a speaker model as the oracle is inspiring and this show a promising way to bridge the gap between the audio-visual navigation task and the vision-language navigation task.

The paper is clearly written and easy to understand. The experiments and analysis are also pretty sufficient.

I expected the success rate and SPL improvement be higher than just a couple percents. I wonder if authors have answer for that.

---

> ### Author Response · Authors · 2022-08-02
> **Response to Reviewer ob8D**
>
> We sincerely thank the reviewer for the positive feedback and thoughtful comments.
>
> - **Will you release the model and code?:**
> Yes, absolutely. We will be releasing our code and pre-trained models upon acceptance.
>
> - **In the supp video, it looks like a couple successful cases were because of the query of stop. Is it possible most the performance improvement comes from the stop query?:**
> Thanks for this careful point. After an analysis of our results, we found that during inference for the unheard sounds, it is only 4.4% of the episodes that the agent query resulted in 'stop' instruction, suggesting that the performance improvements are not mainly from the agent querying near the goal. Note that to have a query that results in a 'stop' instruction, the agent is required to reach near the goal location, and our results suggest that equipping the agent with \pi_l results in the agent reaching the goal location more successfully, resulting in improved performance.
>
> - **Do authors have an intuition of why the performance improvement is not very big? It seems like the language instructions provide pretty strong guidance for navigation.:**
> Thanks for this insightful question. Navigation based on language instructions is a challenging task in itself since language incorporates strong inductive biases and usually spans large vocabularies; as a result, the action predictions can be extremely noisy, imprecise, and misguiding. As a result, utilizing \pi_l does not scale up to a drastically higher performance. However, the key to improved performance is to identify when to query. Our experiments show that AVLEN is able to identify when to query correctly (Table 2) and thus improve performance.

---

> > ### Comment · Reviewer_ob8D · 2022-08-09
> > **Response to rebuttal**
> >
> > Thank you for your response. My questions have been answered and I'm keeping the initial score.

---

### Official Review · Reviewer_zxx9 · 2022-07-22

**Rating:** 5
**Confidence:** 3
**Soundness:** 3 good
**Presentation:** 4 excellent
**Contribution:** 3 good

**Summary:**

In many scenarios, to effectively navigate in the real-world, agents need to combine a variety of sensory information including audio, visual, language, etc. VL agents navigate based on only vision and language input. More recently, SoundSpaces introduced the task of navigating conditioned on an audio source but without language guidance. This paper presents AVLEN, an embodied agent that leverages audio-visual cues in addition to natural language instructions from an oracle for navigation. The agent is trained using hierarchical RL and consists of one high-level policy for deciding when to query the oracle and two low-level navigation policies conditioned on either audio-visual cues or language instructions. The paper demonstrates that the proposed multimodal agent performs better across several evaluation metrics compared to SOTA models on several challenging sound settings in the AudioGoal task. Several ablations are conducted to analysis the usefulness of querying for language guidance from an oracle, using RL for learning when to query, and robustness of the navigation agent to distracting sounds.

**Questions:**

- The paper claims that AVLEN solves VL navigation as a subtask of the SAVi task. It'd make a stronger argument if the paper also included experiments to support this claim by investigating the performance of the AVLEN agent in traditional VL benchmarks like RxR, CVDN, etc.
- Is there an explanation as to why the random model's performance decreased with a larger budget for number of queries? Also the performance gain with more queries seems a bit negligible. It seems that if an agent had access to more oracle guidance, the performance should increase more linearly.
- Is there a breakdown of SR for different goal categories? Wondering if the SR is pretty similar across all categories of sounds or if there are certain categories that are better than others.

**Limitations:**

The authors point out several limitations of their work. Specifically they mention the assumptions that the agent is able to query anywhere in the environment and a pretrained speaker model is used for generating instructions. In terms of societal impact, the authors acknowledge that language-guided / instruction following agents that interface with humans are increasingly relevant and will soon permeate our everyday lives. They state that such agents facilitate automation of challenging human tasks, but neglect to address the negative aspects of such a trend, e.g. putting people out of work. The authors briefly mention that perhaps the simulated environment the agents are trained in, which are hand selected by humans, may be biasing the agent's behavior. It may also be interesting to (which may be quite far-fetched for the current state of the field) consider the negative consequences of introducing sound as an additional modality and equipping robots with audio-capturing sensors which may be used for nefarious purposes.

**Strengths And Weaknesses:**

Strengths:
- First to combine audio-visual navigation with natural language instruction guidance from oracle. Introduces a navigational agent that can leverage full range of modalities, and not just a subset.
- Paper is well motivated both in terms of the task setting and technical approach.
- The authors report SOTA metrics on the benchmark task across all metrics and compare their method against several strong baselines.
- Good set of ablations, demonstrating that the performance gain is bottlenecked by the speaker model and also how the agent learns to distribute the queries throughout the trajectory unlike the MU baseline

Weaknesses:
- There have already been many prior works focused on solving the individual subproblems of AV navigation and separately agents that can interface with an oracle to ask for help.
- Could use an explanation for why n=4 is used for the number of segments used for producing the oracle instruction. Was it chosen arbitrarily?
- Could also compare against a non-hierarchical baseline that uses a single network fusing together the different modalities and producing action to show that the hierarchical abstraction is indeed helpful
- The ablation with distractor sounds makes sense, but I'd imagine that the language guidance should help to disambiguate the distracting sounds a bit more. It might be challenging with only audio-visual, but language instructions should provide a stronger discerning signal.

Minor:
- L330 - "the agents interaction ability is would come useful". wording
- Typo in equation after L234
- For Figure 3c is this a distribution of queries over multiple rollouts?

---

> ### Author Response · Authors · 2022-08-02
> **Response to Reviewer zxx9**
>
> We thank the reviewer for going through our paper and providing thoughtful comments.
> - **Prior works:** We agree with the reviewer that there have been independent prior works for Audio-visual navigation, as well as works that equip the navigating agent to query for help. However, we are the first to bring both these works under a common framework. Our work is also distinctly different from prior approaches where agents may interact with an oracle to ask for help [9, 26, 27]. We elaborate these differences in L93-71 in the paper.
> - **Explanation for n=4:** Since our navigating agent is exploring Matterport 3D environments under the Soundspaces grid, we approximate/adapt the instruction-trajectory pairs from the Fine-Grained R2R dataset into the Soundspaces grid. The mean length of navigation steps for instructions of the Fine-Grained R2R dataset in the Soundspaces grid is 2.53 steps and the standard deviation is 0.87. Speaker model requires relevant information or observations to generate instruction. However, excessive rollouts result in irrelevant observations. We decided to use n=4 steps after carefully considering these statistics so as to generate instructions such that the Speaker observes crucial information for navigation; however, it does not take too many extra steps/observations to confuse the agent.
> - **Comparison with non-hierarchical baseline:** Indeed, we initially approached the design of our architecture using a single model to work with all three modalities, in a non-hierarchical setting. However, we soon faced technical challenges with its implementation, which resulted in designing our hierarchical model . We list below the key challenges in designing non-hierarchical baseline:
>     - **Mismatch at Inference:** At any time step, if an agent receives a language instruction, it requires to take multiple steps based on the query. However, for audio-visual navigation, it requires taking a single step, and after each step, requires to decide if it needs help from the oracle. As a result, there is a mismatch between model behavior at training and inference, which we couldn’t mitigate with a single model.
>     -  **Complication in Training:** A simple approach to training a joint model is to first either train the model with instructions and then train for the audio-visual navigation task. However, the model suffers from forgetting and does not perform well for the initially trained task. An alternative is to train the model jointly. However, joint training requires the model to roll out specific episodes that match with language-based instruction annotation multiple times and thus turned out to be infeasible.
> - **Performance for distracting sounds:** Even though the performance differences are not much, we did see improvements for the unheard distractor sound compared to unheard sound. Specifically, SR improved on unheard sound (by 1.4%) over SAVi, while we obtained a +2.2% in SR for unheard distractor. If we use the ground truth instructions from the oracle, we obtain +12.6% in SR on unheard distractor sound and +11.9% in SR on unheard sound.
> - **Figure 3c query distribution:** Yes, the distribution of query is over multiple (100) rollouts.
> - **Experiments in VL benchmarks:** Thanks for this point. We did think of this direction quite elaborately. Note that, AVLEN solves VL navigation as a subtask of the SAVi task. However, we cannot directly apply our learned \pi_l to RxR or CVDN dataset evaluation as the grid system for RxR or CVDN is completely different from Soundspaces and thus needs separate training. Instead, we analyze the performance of \pi_l policy in a VLN test-set created by us. Please check section 7 of the Appendix for details.
> - **Performance decrease with a larger budget:** As is explained in our discussions in the paper, the key to our improved performance is to identify when to query. If the agent can decide when the \pi_g is likely to fail and rely on \pi_l instead, it can improve performance. Random or Uniform querying policy cannot use such information, and thus their success does not improve with increased queries. The noisy and imprecise \pi_g when combined with random querying results in more navigation digressions, reducing the success.
> - **Breakdown of SR for goal categories:** We provide a breakdown of the success rate (SR) for different sound categories for heard sound. The SR is mostly similar across different categories except for ‘bathtub’ and ‘fireplace’, for these classes perhaps the sounds are subtle and ambiguous. Category: Accuracy%. cabinet:39.1,  picture:35.4, chair:41.8, counter:53.1, toilet:28.5,  cushion:31.6, plant:40.6, sink:27.5, table:32.9, bed:27.7, stool:50.0, bathtub:0.0, sofa:35.2, tv_monitor:14.2, towel:24.0, shower:50.0, clothes:66.6, chest_of_drawers:23.0, seating:26.3, fireplace:0.0, gym_equipment:66.6
> - **Limitations:** We have added the negative aspects of the automation trend as suggested in the revised version.

---

### Official Review · Reviewer_kLxQ · 2022-07-24

**Rating:** 6
**Confidence:** 3
**Soundness:** 2 fair
**Presentation:** 2 fair
**Contribution:** 3 good

**Summary:**

This paper introduces AVLEN, Audio-Visual-Language Embodied Navigation. In this setup, an embodied agent aims to find the location of an audio event by navigating in a simulated 3D visual world by interacting with the environment and an oracle. The oracle provides free-form natural language utterances to guide the agent. To address this task, a hierarchical RL system is proposed. Query policy \pi_q learns whether to use audio-goal policy \pi_g or query the oracle and use language-feedback policy \pi_l. Experiments show that the proposed method surpasses audio-only methods and successfully learns when to query the oracle.

**Questions:**


* L87 we --> our

* L193 please eliminate epsilon notation for policies.

* L243 First, please clarify the intuition behind this formulation for the reward function. Second please add equation number. Also, please clarify how you choose hyperparameters r_neg and r_f

* L251 Please explain the entropy loss. It is not clear to me.

* Table1: Please add AVLEN only \pi_l where budget is infinity. also add a column with checkmarks for language input available or not. AFAIK none of the models you compare have access to the oracle.

* L278 in front --> in front

* L281-282 How do you ensure that the sound comes from a unique object? There are several objects of the same type in the environment.

* Figure 3a: please report  \pi_g, \pi_l only models here as well.

* Figure 3b: please report K=0,1,2 as well

* L328 it's --> its


**Limitations:**

Please discuss limitations in terms of language i.e. you only worked with English annotations also visual diversity i.e. MatterPort images come from North American houses.

**Strengths And Weaknesses:**

* S1 paper contributes to the literature by showing that audio-only methods can be improved with interactive agents when they are provided with natural language instructions

* S2 paper shows that rather than using the model's uncertainty as the query policy, we can train the system to learn when to query.

* W1 While reporting results, the paper misses some essential information or experiments which could be misleading or hard to evaluate fairly the merit of the work.

---

> ### Author Response · Authors · 2022-08-02
> **Response to Reviewer kLxQ**
>
> We thank the reviewer for going through our paper and providing thoughtful comments.
> - **Typos/Clarifications/Experiments:**
> Thanks for pointing out the typos and grammatical issues in the paper. We have included checkmarks and experiments in the paper as suggested. Further, we have also fixed the typos pointed out and added equation numbers. The remaining typos will be fixed in the final paper.
>
> - **First, please clarify the intuition behind this formulation for the reward function:**
> Our reward formulation is inspired by the well-established distance-to-goal reward signal typically used in embodied navigation tasks [5,6], however, has been adapted to our problem setting. Specifically, at any time step, either the Audio Goal policy \pi_g or the Language policy \pi_l takes a step in the environment and collects rewards based on the distance-to-goal. Since our formulation requires a budgeted querying policy, we use two negative rewarding functions so that the agent learns to query within the budget, as well as to query economically. Concretely, (i) the negative reward \zeta_q limits the agent from querying all the time, while (ii) the negative reward \zeta_f discourages the agent from querying within \tau consecutive steps from a previous query step. Since, we formulate soft constraints to incorporate the budget (i.e., on the allowed number of queries) in our formulation, the hyperparameter \r_neg in \zeta_q is designed such that \zeta_q increases only linearly until the allowed number of queries is reached by the agent, beyond which the \zeta_q increases exponentially. From experiments, we found that such a triaged negative penalty design improves learning this budget softly against a single penalty function (e.g., linear alone or exponential alone). As for \zeta_f, this negative reward linearly penalizes using a reward value \r_f if the time difference between two successive queries is low (i.e., queries too frequent), thereby making the agent learn querying economically.
> - **Please clarify how you choose hyperparameters r_neg and r_f:**
> The selection of \r_neg and \r_f were done via cross-validation. Further, the selection of the value of negative rewards was done in a way so that the reward contributions from \pi_g and \pi_l are close to each other.
> - **Please explain the entropy loss:**
> The entropy loss is used to encourage exploration. The key idea is to update the weights in a way that increases the entropy of model prediction of \pi_g during training. It enables the agent to explore the environment more by sampling diverse actions.
> - **Please add AVLEN only \pi_l where the budget is infinity. :**
> Thanks for this point. Please find below the result using AVLEN using only \pi_l. Note that this result is obtained from a model that is trained using \pi_l and \pi_g, however at test time, we used only \pi_l for navigation. The lower performance of this experiment suggests that our model shares the audio-visual cues both from \pi_g and \pi_l for navigation (as was done during training), and thus using only \pi_l at evaluation (even with an infinite budget) is perhaps sub-optimal. As training the model using only \pi_l is time-consuming, we will include those results in the final paper.
>     For Unheard sound:
>     |      Approac| Success | SPL | SNA | DTG | SWS |
>     | :-: | :-: |:-:|:-:|:-:|:-:|
>     | AVLEN (only \pi_l) | 9.0|6.0|6.0|15.2|4.0|
> - **How do you ensure that the sound comes from a unique object? There are several objects of the same type in the environment:**
> The training and testing episodes available with the dataset include information on which particular object in the environment is considered for navigation and localization. From the perspective of an agent, it estimates the location and category of the sounding object which distinguishes it from other similar types of objects. The agent learns to estimate location and object category in a supervised manner (i.e., offline training for object category detection based on the sound and online training of location estimation based on the sound) where the input is a two-channel spectrogram.
> - **Please report \pi_g, \pi_l only models here as well:**
>  We have added AVLEN (Only \pi_g) and AVLEN (GT) in Figure 3a similar to Table 1 in the revised paper. We skip the results on \pi_l in the paper for now, as the complete results will need further training of our models (as discussed above), and promise to include them in the final paper.
>     For distractor sound:
>     |     Approach       | Success | SPL | SNA | DTG | SWS |
>     | :-:                |   :-:   | :-: | :-: | :-: | :-: |
>     | AVLEN (only \pi_g) | 11.8    | 7.4 | 5.0 | 13.1| 8.4 |
>     | AVLEN (GT)         | 24.4    | 15.3| 11.3| 11.3| 21.5|
> - **Please report K=0,1,2 as well:**
> Figure 3b has been updated to include K=0,1,2 as well.
> - **Limitations and Societal Impacts:**
> Thanks for this important point. We have added it in the revised version.

---

### Meta-Review · Area_Chair_rRBa · 2022-08-31

**Recommendation:** Accept
**Confidence:** Certain

**Metareview:**

The authors present a model for soundscapes that is able to request guidance from an oracle. This requires that the agent know when to query the user (oracle) for help. This sits somewhere between the work on oracle guidance and language instruction following.  Ablations and comparisons are provided but do not fully explore the natural set of questions about why/where the model performs best.  Note that this oracle deviates from that in the most relevant prior work by Nguyen and these instructions differ from those generated by a model in papers by Thomason.

Overall, the work makes a nice contribution to bridging existing literatures within EAI.

**Award:**

No

---

### Decision · Program_Chairs · 2022-09-14

Accept